# Health Related Behaviors and Life Satisfaction in Patients Undergoing Infertility Treatment

**DOI:** 10.3390/ijerph19159188

**Published:** 2022-07-27

**Authors:** Małgorzata Nagórska, Małgorzata Lesińska-Sawicka, Bogdan Obrzut, Dariusz Ulman, Dorota Darmochwał-Kolarz, Barbara Zych

**Affiliations:** 1Institute of Medical Sciences, Medical College of Rzeszow University, 35-959 Rzeszow, Poland; bogdan.obrzut@gmail.com (B.O.); ddarmochwal@ur.edu.pl (D.D.-K.); 2Department of Nursing, Stanislaw Staszic State University of Applied Science in Pila, 64-920 Pila, Poland; mlesinska@puss.pila.pl; 3Department of Obstetrics and Gynaecology, Pro-Familia Hospital, 35-001 Rzeszow, Poland; ulmandar@gmail.com; 4Institute of Health Sciences, Medical College of Rzeszow University, 35-959 Rzeszow, Poland; ba.zyc@wp.pl

**Keywords:** infertility, health related behaviors, life satisfaction, lifestyle

## Abstract

The aim of the study was to evaluate the level of life satisfaction and health behaviors presented by patients with diagnosed infertility. This cross-sectional study included 456 patients (235 women and 221 men) from infertile couples in southeastern Poland from June 2019 to February 2020. Participants completed a questionnaire on sociodemographic characteristics, the Health Behaviors Inventory (HBI), and the Satisfaction with Life Scale (SWLS). The average score of severity of health behaviors for the study group was 82.96 points. Satisfaction with life at a higher level was declared by 57.6% of respondents, at an average level was declared by 31.4%, and at a lower level was declared by 11%. The SWLS score for the entire study group was 24.11 points (6.82 points on the sten scale). Respondents who achieved a higher rate of life satisfaction also had a higher level of severity of health behaviors (*p* < 0.0001). There were no significant differences between male and female SWLS scores, although the women had significantly higher rates of severity of health behaviors than men. The level of health behavior is positively related to life satisfaction in infertile patients. Medical personnel should conduct health education on a healthy lifestyle that promotes the improvement of reproductive health.

## 1. Introduction

Infertility is becoming a growing health problem. According to the European Society for Human Reproduction and Embryology (ESHRE), one in six couples worldwide suffers from infertility [1]. The World Health Organization’s (WHO) clinical definition of infertility refers to infertility as “a disease of the reproductive system defined by failure to achieve a clinical pregnancy after 12 months or more of regular unprotected sexual intercourse” [2]. The WHO’s epidemiological definition describes infertility as “women of childbearing potential at risk of becoming pregnant who report unsuccessful attempts to become pregnant for more than 2 years” [3]. The American College of Obstetricians and Gynecologists (ACOG) and the American Society for Reproductive Medicine (ASRM) define infertility as failure to achieve pregnancy within 12 months of unprotected intercourse or therapeutic donor insemination in women <35 years or within 6 months in women >35 years [4].

Infertility is a very specific disease, because, in the physical sense, it does not cause pain and does not lead to disability, and also does not threaten human life. This disease often has the characteristics of a personal failure and affects the mental sphere, which often causes ailments worse than physical pain. The lack of children in a relationship can lead to disorganization in the functioning of people who want to become parents, causing them to focus all their attention on the problem of getting pregnant and to take health behaviors that can make it easier to become a parent [5].

Health behaviors are any behaviors of an individual that are part of everyday functioning, affecting the state of their health. These activities are based on the practical use of existing knowledge about health and diseases. They are the subject of relatively free personal choices and decisions [6].

Health behaviors are reactions to all health-related situations, as well as habits and intentional activities. Each person makes permanent, intentional, conscious, and independent choices of behavior that have a positive or negative impact on health. Health behaviors that have a positive impact on human health, in addition to self-control of the body, responsibility for one’s own health, and a positive attitude, are primarily a healthy diet, regular physical activity, and the optimal amount of sleep per day. Risk factors for diseases and, thus, behaviors that negatively affect health are smoking, improper diet, low or no physical activity, and alcohol abuse [7].

Health, in every sphere of functioning, is associated with life satisfaction [8]. Infertility as a health dysfunction may be the cause of a decrease in life satisfaction as a result of the existence of psychosocial disorders, such as increased stress related to infertility treatment, affective disorders, partner problems, and lack of social support or social exclusion [9]. Infertility treatment is a long-term process, spread over months or even years, not giving a guarantee of success. It generates a constant state of uncertainty, which becomes a chronic state of psychological discomfort. Each stage of treatment can become a source of further trouble. Women who choose to treat infertility are exposed to many negative feelings about various aspects of their lives and overall life satisfaction [10].

Life satisfaction is defined as an assessment of feelings and attitudes about a person’s life at a particular time that ranges from negative to positive, and it is a cognitive, judgmental process based on a comparison of individual circumstances to an appropriate standard [11]. People seeking medical help for infertility show lower levels of life satisfaction [12].

Partners’ health behaviors and overall life satisfaction during the diagnosis and treatment of infertility can be important factors in making it easier for potential parents to undergo medical procedures and rigors. Several studies found that there are gender differences in experiencing infertility [13,14,15,16,17,18,19,20].

The aim of the study was to evaluate the level of life satisfaction and health behaviors presented by patients with diagnosed infertility.

## 2. Materials and Methods

The study design was a cross-sectional descriptive study. Data were collected from June 2019 to February 2020, among randomly selected, infertile patients of three gynecological outpatient clinics in southeastern Poland. The presented study is part of a bigger project “Psychosocial problems of patients treated for infertility”.

### 2.1. Participants

Criteria for selection in the study were adult patients (>18 years) with diagnosed infertility according to WHO clinical definition, voluntary consent to participate in the study, and no communication problems. Exclusion criteria were patients <18 years, who did not meet the criteria for the WHO clinical definition of infertility, who had difficulties understanding the language, or who did not agree to participate in the study. To calculate the sample size of the study, the G*Power 3.1.9.2 program (Faul, F., Erdfelder, E., Lang, A.-G., Buchner, A., Düsseldorf, Germany) was used.

### 2.2. Ethical Consideration

The study was conducted in accordance with the Declaration of Helsinki for medical research. The project received a positive opinion from the Bioethics Committee at the University of Rzeszow, Poland (Resolution No. 2018/04/03).

### 2.3. Tools

In the paper-and-pencil study, we used three measurement tools: personal Information form (PIF), Health Behaviors Inventory (HBI), and Satisfaction with Life Scale (SWLS).

#### 2.3.1. Personal Information Form (PIF)

To assess the characteristics of the study group, a survey developed by the first author was used. It consisted of questions, which were focused on sociodemographic data (gender, age, place of living, the level of education, place of residence, and duration of relationship), the duration of treatment of participants, the reason and type of infertility, the duration of attempt to conceive, and the history of assisted reproductive techniques (ART) used.

#### 2.3.2. Health Behaviors Inventory (HBI)

The level of health behaviors of respondents was determined by the Health Behaviors Inventory scale proposed by Zygfryd Juczynski [20]. The scale is addressed to adults to determine health-related behaviors. The scale consists of 24 statements, divided into four subscales: (1) Correct eating habits (CEH) (diet), (2) preventive behaviors (PB) (health recommendations), (3) positive mental attitude (PMA) (stress avoidance), and (4) health practices (HP). Respondents graded every statement on a five-point Likert scale (from 1—almost never to 5—almost always) as how often over the past year they adhered to specific behaviors. On the basis of the responses obtained, the general health behavior index (GHBI) was determined, as well as the intensity of health behaviors in the four categories mentioned above. The value of the health behaviors can range from 24 to 120 points. The overall result is interpreted after the points are converted to a sten scale. A score of 7–10 sten is defined as high, of 5–6 is defined as medium, and of 1–4 is defined as low, which corresponds to areas of 33% for the highest scores and lowest scores on the scale. Indicators for the four subscales were calculated as the average number of points obtained in each. Cronbach’s alpha reliability index was satisfactory, amounting to 0.85 [21].

#### 2.3.3. The Satisfaction with Life Scale (SWLS)

The Satisfaction with Life Scale (SWLS) by Diener, Emmons, Larson, and Griffin (1985) was used to assess the level of life satisfaction [22]. The SWLS is one of the most popular tools for life satisfaction surveys and consists of five statements to which the respondent addresses on a seven-point Likert scale (from 1—I strongly disagree, to 7—I strongly agree). The respondent assesses to what extent each statement refers to their lifestyle. The result of the measurement is an overall indicator of the sense of satisfaction with life. The points are added, and the obtained results in the range from 5 to 35 determine the degree of satisfaction with life.

The degree of life satisfaction is determined by adding up the points (range 5–35), which are transformed to a sten scale (range 1–10). A score of 1–4 sten is interpreted as low, of 5–6 is interpreted as medium, and of 7–10 is interpreted high, which corresponds to areas of 33% for the lowest scores and highest scores on the scale. The scale was adopted to Polish conditions by Zygfryd Juczynski (2001). Cronbach’s alpha reliability index was satisfactory, amounting to 0.81 [21].

### 2.4. Course of the Study

Prior to the implementation of the study, the medical facilities approved its design. A total of 500 patients meeting the inclusion criteria were invited to participate in the survey after their appointment with a gynecologist in three gynecological outpatient clinics. The respondents were informed about the aim of the study and their anonymity, and they were made aware of the possibility of withdrawing from the study at any stage without any consequences. It took an average of 20 min to complete the questionnaire. Each participant could ask additional questions during the research. From the 472 replies, 456 (91%) correctly and fully completed questionnaires were used for statistical analysis.

### 2.5. Statistical Analysis

Statistical analysis was performed using the program IBM SPSS Statistics 20 (SPSS Inc., Chicago, IL, USA). The following estimation and statistical methods were used: the results obtained were subjected to statistical analysis using descriptive statistic methods according to the arithmetic mean (M) and standard deviation (SD). The Pearson χ^2^ independence test and the Mann–Whitney test were used to verify differences between variables measured on the qualitative scale. The differences between quantitative variables were tested using the *t*-test for independent samples. Normality of distributions was examined by the Kolmogorov–Smirnov test; the assumption of homogeneity of variance was previously examined with Levene’s test. A value of *p* < 0.05 was assumed to be statistically significant.

## 3. Results

The study involved 456 patients treated for infertility: 235 women (51.5%) and 221 men (48.5%). The average age of respondents was 33.85 years (range 24–52). The mean age of the women was 33.10 ± 4.33 years, while the mean age of the men was 34.64 ± 5.07 years. Over half of respondents lived in an urban environment (54.4%), while 45.6% lived in rural areas. Most of surveyed patients had a university level of education (64.7%). The average duration of the relationship was 9.14 ± 4.23 years. Couples were most commonly in a relationship between 6 and 9 years (39.7%). The mean time of trying to conceive in the study group was 3.99 ± 2.46 years (range 1–15). Primary infertility concerned most of patients (83.1%), with the remainder affected by secondary infertility (16.9%). Half of the surveyed participants (n = 232, 50.9%) were aware of the medical reason for infertility. In that group, 61.6% (n = 143) indicated the female factor, while 25% indicated the male factor (n = 58). The most commonly used assisted reproduction technique (ART) in patients was intrauterine insemination (IUI) (n = 134, 29.4%). Classical in vitro fertilization (IVF) (n = 48, 10.5%), IVF with micromanipulation (n = 38, 8.4%), or other methods (n = 7, 10.5%) were used less frequently. For 50.2% of the subjects (n = 229), no assisted reproduction technique has been used so far (Table 1).

Analyzing health behaviors and the level of life satisfaction, we calculated the results for the entire study group and made comparisons of results between women and men. The average score of the intensity of health behaviors for the entire surveyed group was 82.96 points. The results within the four areas of health behavior were similar (about 3.50 points). The intensity of health behaviors in relation to sten norms for the majority of respondents was average (46.5%). A high intensity of health behaviors was presented by 30.3%, whereas a low intensity was presented by 23.2% of respondents (Table 2).

In turn, the SWLS score for the entire study group was 24.11 points (6.82 points after conversion to the sten scale). Satisfaction with life at a higher level was declared by 57.6% of respondents, at an average level was declared by one-third of respondents (31.4%), and at a lower level was declared by 11% of respondents (Table 3).

At a later stage of the analysis, we compared the levels of health behaviors and satisfaction with life by gender.

The severity of health behaviors differed significantly between men and women (*p* = 0.0007). Women were more likely to achieve a high rate of severity of health behaviors (37.9%), whereas men were more likely to achieve a low rate (28.1%) (Table 4).

The women presented significantly higher rates of intensity of health behaviors in each of the four dimensions. The smallest, but still significant differences concerned positive mental attitude (*p* = 0.0151) between women (3.61 points) and men (3.48 points). The overall rate of intensity of health behaviors in quantitative terms (scale 24–120 points) was also significantly higher among women (87.94 points) than among men (77.67 points) (*p* < 0.0001) (Table 5).

According to the analysis, the gender of the respondents did not significantly affect life satisfaction, although women were slightly more likely to obtain low scores 14.0%, with men having an average result of 33.5% (Table 6).

There were no significant differences between SWLS scores on a scale of 5–35 (23.80 women vs. men 24.43; *p* = 0.2559); similarly, no significant differences between gender and SWLS were measured on a scale of 1–10 points (6.71 women vs. 6.93 men; *p* = 0.2565) (Table 7).

A statistically significant relationship between the severity of health behaviors and life satisfaction was demonstrated. Those who achieved a higher rate of life satisfaction also had higher levels of severity of health behaviors (*p* < 0.0001) (Table 8).

## 4. Discussion

This study aimed to evaluate the level of life satisfaction and health behaviors presented by patients with diagnosed infertility.

Many authors have emphasized the importance of a healthy lifestyle for both general and reproductive health. Among the factors affecting fertility, the most commonly mentioned are nutrition and maintenance of proper body weight, stimulants and medications, physical activity, stress, sleep and leisure, environmental pollution, occupational exposures, and age during the decision to procreate [23,24,25,26,27,28,29].

To determine the lifestyle and health behaviors of our respondents, we used the HBI scale. However, we did not find studies among infertile patients using this scale, which is why we refer to similar studies using other research tools in the discussion.

Our study indicated that the men had a significantly lower level of health behaviors than women in each of the four dimensions discussed: CEH, PB, PMA, and HP.

The nutritional status and maintenance of normal body weight in both women and men are indicated as important factors that may affect fertility [27,29,30,31,32]. A proper, balanced diet serves good health in general, but there are also other factors with a big impact on reproductive health [25]. Overweight and obese patients are advised to reduce their BMI, and those who are underweight are recommended to gain weight [27,33]. Maintaining proper body weight has a positive effect on the hormonal balance, which translates into effective treatment outcomes [33].

Infertility treatment requires substantial time and discipline in completing procedures that cannot be postponed in time. According to The Cardiff Fertility Studies, the chance of achieving the pregnancy is around 70% when patients follow the treatment recommendations [34].

Our study confirmed gender differences in treatment approaches and that men are less disciplined than women in adherence to medical recommendations and in participation in therapy.

This was also confirmed by the results of other authors, where it was shown in the case of problems related to reproductive health that only every 10th man goes to the doctor, while every fifth man decides to visit a specialist only when the disturbing symptoms do not go away [25,35]. Women, on the other hand, had better health, which resulted from more frequent check-ups at the doctor, e.g., during screening tests (PAP smear) [26,36].

The results of many studies confirmed that the diagnosis and treatment of infertility are fraught with considerable stress; therefore, the psychological aspects in the treatment of infertile couples should not be ignored [27,37,38]. A positive attitude has a key role in therapy, and pessimism is treated as a risk factor for IVF treatment failure [39,40].

Sleep continuity disorders can also affect fertility [41], uninterrupted sleep, and circadian rhythms, which may play an important role in the success of infertility treatment [42]. Physical activity also affects both the reproductive capacity and the course of pregnancy and childbirth in women, as well as the reproductive potential of men [43]. However, caution should be exercised with activity, because “overtraining syndrome” can have the opposite effect, especially in men, as it adversely affects sperm parameters [44,45].

The intensity of our respondents’ health behaviors was at an average level. Gormack and Rooney in their studies also confirmed that patients do not fully adhere to the recommended lifestyle modification [46,47]. Since lifestyle is modifiable, it should be carefully analyzed from the beginning of treatment and patients educated in this regard [48].

In our work, we also studied the life satisfaction of infertile patients. In the study group, the average life satisfaction score was at the level of 24.11 points (23.80 women vs. men 24.43). According to our analyses, gender did not significantly affect the perceived life satisfaction, although women were slightly more likely to achieve low scores and men to achieve average results. Navid et al. (2017) (*n* = 248 infertile couples) also did not show significant differences in SWLS between partners (total score: 21.35, male: 24.40, female: 25.51) [12].

The study conducted by Adachi et al. among 449 Japanese patients seeking fertility treatment showed that the mean score of SWLS for women was significantly lower than that for men (women = 21.2 vs. men = 22.4) [49]. In another study conducted on infertile women in Iran, results indicated that women undergoing infertility treatments were quite satisfied with their lives (score 21–25) (which corresponds to a value of 6–7 on a sten scale) [50]. In turn, Hammarberg and coauthors studied the opinions of men from infertile couples and showed the impact of the diagnosis of infertility with the functioning of the relationship and on their level of life satisfaction. Men who believed that the diagnosis of infertility had a negative impact on relationships also showed lower scores on the SWLS scale (negative impact, SWLS score—22.96 points vs. neutral or positive impact, SWLS score—25.95 points) [51]. McQuillan et al. stated that individuals who experienced infertility had a lower life satisfaction, compared to those who did not [13]. 

This study had some limitations. First, the study was conducted only among infertile patients, which limits the comparison with fertile couples. Secondly, this study was conducted in one city and region of Poland, and the results of the study cannot be generalized to the entire society. Thirdly, we do not know the satisfaction with life and health behaviors before the diagnosis of infertility in respondents. In addition, it was difficult to compare and discuss the results, because we did not find other studies among infertile couples where HBI was used. It would also be worth checking the stress and anxiety level according to HBI and SWLS.

## 5. Conclusions

The level of life satisfaction is positively correlated with the intensity of health behaviors in infertile patients. Verification and optimization of health behaviors should be an essential part of infertility therapy. Due to the fact that this is a factor that depends to a large extent on the patients themselves, they should be constantly made aware of this. Medical personnel should conduct health education aimed at promoting a proper lifestyle that improves same general and reproductive health. In some cases, this can also have a positive effect on reducing costly invasive procedures during therapy. It is also worth conducting further studies, referring in more detail to individual areas of the lifestyle of infertile patients.

## Figures and Tables

**Table 1 ijerph-19-09188-t001:** Characteristics of the respondents.

	n	%
**Gender**		
Female	235	51.5
Male	221	48.5
**Age (years)**		
24–29	89	19.5
30–34	170	37.3
35–39	144	31.6
40 and more	53	11.6
**Place of residence**		
Urban	248	54.4
Rural	208	45.6
**Level of education**		
Primary	3	0.7
Vocational	34	7.5
Secondary	124	27.2
Higher	295	64.7
**Duration of the relationship (years)**		
To 5	87	19.1
6–9	181	39.7
10–14	128	28.1
15 and more	60	13.2
**Duration of attempts to conceive (years)**		
1–2	150	32.9
3–4	159	34.9
5–6	89	19.5
7 and more	58	12.7
**Infertility type**		
Primary	379	83.1
Secondary	77	16.9
**Reason of infertility**		
Known	232	50.9
Unknown	224	49.1
**ART used so far**		
IUI	134	29.4
IVF	48	10.5
IVF ICSI	38	8.4
Other	7	1.5
None	273	50.2

ART—assisted reproductive technique, IUI—intrauterine insemination, IVF—in vitro fertilization, ICSI—intracytoplasmic sperm injection.

**Table 2 ijerph-19-09188-t002:** Intensity of health behaviors in points and sten norm.

				**Intensity of Health Behaviors (Points)**			
		**CEH**	**PB**	**PMA**		**HP**	**Intensity of Health Behaviors (Points)**
M		3.35	3.52	3.55		3.41	82.96
SD		0.73	0.72	0.60		0.64	12.69
Min.		1.33	1.67	1.33		1.17	43.00
Max.		5.00	5.00	5.00		5.00	115.00
Percentile	25	2.83	3.00	3.17		3.00	75.00
	50	3.33	3.50	3.50		3.50	83.00
	75	3.83	4.00	4.00		3.83	92.00
**Intensity of Health Behaviors (Sten Norms)**							
		**Frequency**		**Percentage**	**Percentage of valid**		**Cumulative percentage**
Valid	Low	106		23.2	23.2		23.2
	Medium	212		46.5	46.5		69.7
	High	138		30.3	30.3		100.0
	Total	456		100.0	100.0		

M—arithmetic mean, SD—standard deviation; CEH—correct eating habits, PB—preventive behaviors, PMA—positive mental attitude, HP—health practices.

**Table 3 ijerph-19-09188-t003:** Scale of life satisfaction—raw results (5–35) and on the sten scale (1–10).

		**SWLS (5–35)**			**SWLS Sten (1–10)**
M		24.11			6.82
SD		5.26			1.91
Min.		5			1
Max.		35			10
Percentile	25	21.00			6.00
	50	25.00			7.00
	75	28.00			8.00
**The Scale of Life Satisfaction (Sten Norms)**					
		**Frequency**	**Percentage**	**Percentage of valid**	**Cumulative percentage**
Valid	Low	50	11.0	11.0	11.0
	Medium	143	31.4	31.4	42.3
	High	263	57.6	57.6	100.0
	Total	456	100.0	100.0	

M—arithmetic mean, SD—standard deviation.

**Table 4 ijerph-19-09188-t004:** Intensity of health behaviors and gender.

			Gender		Total
			Women	Men	
Intensity of health behaviors	Low	*n*	44	62	106
		%	18.7%	28.1%	23.2%
	Medium	*n*	102	110	212
		%	43.4%	49.8%	46.5%
	High	*n*	89	49	138
		%	37.9%	22.2%	30.3%
Total		*n*	235	221	456
		%	100.0%	100.0%	100.0%
χ^2^ = 14.537; *p* = 0.0007 *					

* Pearson χ^2^ test of independence.

**Table 5 ijerph-19-09188-t005:** Intensity of health behaviors (in detail) and gender.

Gender		CEH	PB	PMA	HP	GHBI (24–120)
Women	M	3.59	3.83	3.61	3.62	87.94
	Me	3.67	3.83	3.67	3.67	88.00
	Min.	1.50	2.33	2.17	2.00	49.00
	Max.	5.00	5.00	4.83	5.00	115.00
	SD	0.67	0.57	0.58	0.58	10.96
Men	M	3.10	3.18	3.48	3.19	77.67
	Me	3.00	3.17	3.50	3.17	78.00
	Min.	1.33	1.67	1.33	1.17	43.00
	Max.	5.00	4.67	5.00	5.00	110.00
	SD	0.71	0.71	0.62	0.64	12.29
Total	M	3.35	3.52	3.55	3.41	82.96
	Me	3.33	3.50	3.50	3.50	83.00
	Min.	1.33	1.67	1.33	1.17	43.00
	Max.	5.00	5.00	5.00	5.00	115.00
	SD	073	0.72	0.60	0.64	12.69
*p* *		<0.0001	<0.0001	0.0151	<0.0001	<0.0001

* *t*-Test for independent tests (normality of distributions was examined by the Kolmogorov—Smirnov test. The results did not differ significantly from the normal distribution; hence, this test was used, after previously examining the assumption of homogeneity of variance with the Levene test). M—arithmetic mean, Me—median, SD—standard deviation CEH—correct eating habits, PB—preventive behaviors, PMA—positive mental attitude, HP—health practices, GHBI—general health behaviors index.

**Table 6 ijerph-19-09188-t006:** The Satisfaction with Life Scale and gender.

Gender			Gender		Total
			Women	Men	
Satisfaction with Life Scale (SWLS)	Low	*n*	33	17	50
		%	14.0%	7.7%	11.0%
	Medium	*n*	69	74	143
		%	29.4%	33.5%	31.4%
	High	*n*	133	130	263
		%	56.6%	58.8%	57.7%
Total		*n*	235	221	456
		%	100.0%	100.0%	100.0%
χ^2^ = 4.904; *p* = 0.0861 *					

* Pearson χ^2^ independence test.

**Table 7 ijerph-19-09188-t007:** The Satisfaction with Life Scale—raw results (5–35) and on the sten scale (1–10) and gender.

		Women (*n* = 235)		Men (*n* = 221)		Total (*n* = 456)	
		SWLS (5–35)	SWLS (1–10)	SWLS (5–35)	SWLS (1–10)	SWLS (5–35)	SWLS (1–10)
M		23.80	6.71	24.43	6.93	24.11	6.82
SD		5.43	1.95	5.05	1.87	5.26	1.91
Min.		5	1	7	1	5	1
Max.		35	10	35	10	35	10
Percentile	Q1	20	5	21	6	21	6
	Q2 (median)	24	7	25	7	25	7
	Q3	28	8	28	8	28	8

M—arithmetic mean, SD—standard deviation.

**Table 8 ijerph-19-09188-t008:** Intensity of health behaviors and life satisfaction.

			Satisfaction with Life Scale (SWLS)			Total
			Low	Medium	High	
Intensity of health behaviors	Low	*n*	23	44	39	106
		%	46.0%	30.8%	14.8%	23.2%
	Medium	*n*	18	73	121	212
		%	36.0%	51.0%	46.0%	46.5%
	High	*n*	9	26	103	138
		%	18.0%	18.2%	39.2%	30.3%
Total		*n*	50	143	263	456
		%	100.0%	100.0%	100,.0%	100.0%
χ^2^ = 40.735; *p* < 0.0001 *						

* Pearson χ^2^ independence test.

## Data Availability

The data analyzed in the study are available upon request to the corresponding author.

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
