# Peer review of "Health Related Behaviors and Life Satisfaction in Patients Undergoing Infertility Treatment"

_ijerph, 2022, doi:10.3390/ijerph19159188_

Round 1

Reviewer 1 Report

the authors accurately followed reviewers' comments and definitively improved the quality of the manuscript, which is now fully suitable for the publication in the Journal. 

Reviewer 2 Report

Authors answered all the points mentioned previously. 

This manuscript is a resubmission of an earlier submission. The following is a list of the peer review reports and author responses from that submission.

Round 1

Reviewer 1 Report

I have carefully revised the manuscript entitled “Health Related Behaviors and Life Satisfaction in patients undergoing infertility treatment” by Nagórska and colleagues. This article aims to investigate the level of life satisfaction and of health behaviors in patients with a diagnosis of infertility, using two scales and a questionnaire. Despite the interesting aim, the manuscript, as well as the study, presents several major flaws that make it not suitable for the publication in the journal in its current form. I thus recommend a rejection with the possibility of resubmission after an accurate work of rewriting by the authors.   

The main flaw regards the presentation of the two used tools and the absence of the questionnaire. The two scales are almost incomprehensible, I suggest to rewrite the methods section in order to clearly explain them. Some examples:

-          Is the GHBI the same of the HB? In the text it is impossible to distinguish.

-          Lines 71-81: the calculation used to obtain the scores is not at all clear.

-          The presentation (or maybe the choice) of the inclusion and exclusion criteria is too simple.

-          The study period differs in relation to that presented in the abstract.

-          The questionnaire is absent, it is not conceivable considering that is a tool of the study.

In the results, and also throughout all the manuscript, the authors use the word “severity”, but it is not clear if they used it in the right meaning. The term “severity” has undoubtable a negative sense, but in the text it is not clear if it is the sense the author want to use. This problem is mainly due to the incomprehensible presentation of the HBI, from which it is not possible to distinguish a positive behavior from a negative one and thus a positive behavioral score from a negative one.

Throughout the text the are other flaws that seem to show the lack of care during the writing process:

-          In the abstract, the sum of the percentages is not 100% (line 17).

-          The study period differs from the abstract to the text.

-          In table 6 and 8 there are word not translated in English.

-          Globally, the English is quite poor and does not help the comprehension of the text.

Reviewer 2 Report

The subject is interesting due to the real impact of health-related behaviors on infertility.

Comments and suggestions for the authors

-       Define the target population with infertility because there are some differences between age subcategories.

-       I did not notice the application of the Strobe checklist criteria

-       Please move table 1 below the participants' section

-       Table 1 correct higer in higher

-       Line 154 translated into English

-       Table 8 translated into English

-       What results were obtained after the recommended lifestyle change?

-       How many patients underwent ART?

-       After completing the demographic questionnaire, infertile patients should have completed a scale of anxiety and depression. How would this scale change the outcome of the study? To be entered in the limitations section.

Reviewer 3 Report

Although it is a very important issue and it is presented adequately by the authors, there are some points need improvement

1. Abstract should have the form "introduction-materials methods-results-conclusion"

2. English language needs improvement

3. Authors do not mention if questionaires standarized for the population and what did they do if someone could not understand

4. Exclusion criteria should be more detailed

5. Authors should mention the area where population study answered the questions (home, clinic, hospital area etc.)

6. In my opinion tables are very extended and should be incorporated

7. Authors should write a paragraph reffering to limitations of thei study

8. Authors should compare their results with previous studies and explain why they used different materials